# Distribution of high-dimensional entanglement via an intra-city free-space link

Fabian Steinlechner[1,*], Sebastian Ecker[1,*], Matthias Fink[1], Bo Liu[1,2], Jessica Bavaresco[1], Marcus Huber[1], Thomas Scheidl[1] & Rupert Ursin[1,3]

Quantum entanglement is a fundamental resource in quantum information processing and its distribution between distant parties is a key challenge in quantum communications. Increasing the dimensionality of entanglement has been shown to improve robustness and channel capacities in secure quantum communications. Here we report on the distribution of genuine high-dimensional entanglement via a 1.2-km-long free-space link across Vienna. We exploit hyperentanglement, that is, simultaneous entanglement in polarization and energy-time bases, to encode quantum information, and observe high-visibility interference for successive correlation measurements in each degree of freedom. These visibilities impose lower bounds on entanglement in each subspace individually and certify four-dimensional entanglement for the hyperentangled system. The high-fidelity transmission of high-dimensional entanglement under real-world atmospheric link conditions represents an important step towards long-distance quantum communications with more complex quantum systems and the implementation of advanced quantum experiments with satellite links.

[1] Institute for Quantum Optics and Quantum Information (IQOQI), Austrian Academy of Sciences, Boltzmanngasse 3, A-1090 Vienna, Austria. [2] School of Computer, NUDT, Changsha 410073, China. [3] Vienna Center for Quantum Science and Technology (VCQ), Faculty of Physics, University of Vienna, Boltzmanngasse 5, A-1090 Vienna, Austria. * These authors contributed equally to this work. Correspondence and requests for materials should be addressed to F.S. (email: fabian.steinlechner@oeaw.ac.at) or to R.U. (email: rupert.ursin@oeaw.ac.at).

The distribution of quantum entanglement between distant parties is one of the main technological challenges in the pursuit of a global-scale quantum Internet. Several proof-of-concept studies have already demonstrated high-fidelity transmission of photonic entanglement via terrestrial long-distance free-space links[1–3], and established the viability of employing optical satellite links for quantum communication on a global scale[4,5], and beyond[6]. However, until very recently, these experimental studies have been focused on bipartite binary photonic systems, that is, the simplest state space that can exhibit quantum entanglement. Specifically, polarization qubits have been the system of choice for free-space quantum communications for over a decade.

Encoding several qubits per transmitted photon increases channel capacity and yields significant benefits in the implementation of advanced quantum information processing protocols, such as improving resilience with respect to noise and eavesdropping in secure quantum communications[7–14]. Hence, increasing the dimensionality of entangled quantum systems can be considered one of the next key technological steps towards the realization of more practical quantum information processing protocols in real-world scenarios. Furthermore, from a fundamental physics point of view, the more diverse variations of non-classical correlations that are possible in a large state space also provide a platform for diverse quantum physics experiments[15–18].

High-dimensional quantum information can be encoded in various photonic degrees of freedom (DOF), such as transverse orbital angular momentum (OAM)[19–22], discrete photon arrival time bins[23] or continuous-variable energy–time modes[24,25]. The transmission of classical OAM modes through turbulent atmosphere has been studied in several field trials[26,27] and OAM multiplexing has already been used to achieve record channel capacity in free-space optical communications[28]. While OAM entanglement has already been successfully demonstrated after atmospheric propagation[29], active wavefront correction will be required to fully exploit the potential of OAM encoding. The development of suitable adaptive optics systems is an immensely challenging field of ongoing research. Energy–time entanglement and its discrete analogue time-bin entanglement both offer alternatives for high-dimensional state encoding. Time-bin qubits[30,31] have been routinely used in fibre-based quantum key distribution systems, which has culminated in the recent demonstration of quantum teleportation over long-distance fibre links[32,33] but have only recently been considered as a viable option for free-space quantum communications in presence of atmospheric turbulence[34,35].

The dimensionality of the state space can also be increased by simultaneously encoding quantum information in several DOF[36]. This has the significant advantage that single-photon two-qubit operations can be implemented deterministically between different DOF using only passive linear optics devices[37,38]. Furthermore, simultaneous entanglement in multiple DOF, known as hyperentanglement[39], is readily engineered via the process of spontaneous parametric down-conversion (SPDC) in nonlinear crystals[40]. Hyperentanglement has been exploited in the realization of numerous advanced experiments, such as hyperentanglement-assisted Bell-state measurements[3,41–43], quantum teleportation of multiple DOF of a single photon[44,45], robust quantum communications with increased channel capacity[46] and efficient entanglement purification schemes[47–49]. However, experiments that exploit hyperentanglement have not yet ventured beyond the distance limitations of optical tables and protected laboratory environments.

In this article, we report on the distribution of energy–time and polarization hyperentangled photons via a 1.2-km-long intra-city free-space link. We observe high-visibility two-photon interference

for successive correlation measurements in the polarization space and a two-dimensional energy–time subspace and certify four-dimensional entanglement in the combined system. Our assessment of energy–time entanglement is based on the observation of Franson interference in unbalanced polarization interferometers[50,51]. This simple approach is highly suitable for the exploitation of such states in future quantum experiments with satellite links.

## Results

**Experimental set-up.** The experiment (depicted in Fig. 1) was performed with an ultra-bright source of hyperentangled photons and a detection module (Alice) located at the Institute for Quantum Optics and Quantum Information (IQOQI) and a receiver station (Bob) at the University of Natural Resources and Life Sciences (BOKU) in Vienna.

The source of hyperentangled photons (see 'Methods' section) was based on type-0 SPDC in a polarization Sagnac interferometer[52,53] with a continous-wave 405-nm pump laser. It produced fibre-coupled polarization-entangled photon pairs with a two-photon coherence time $t_c \lesssim 1\,\mathrm{ps}$ and centre wavelengths $\lambda_A \sim 840\,\mathrm{nm}$ and $\lambda_B \sim 780\,\mathrm{nm}$, where subscripts A and B label the respective single-mode fibre for Alice and Bob. Since the emission time of a photon pair is uncertain within the significantly longer coherence time of the pump laser ($t_p \gtrsim 100\,\mathrm{ns}$, photons A and B were entangled in energy–time[50]. In our proof-of-concept demonstration, we focused on a two-dimensional subspace of the high-dimensional energy–time space (see 'Methods' section). The total state space considered in our proof-of-concept experiment is thus a four-dimensional hyperentangled state in polarization and energy–time:

$$
\begin{aligned}
|\Psi\rangle_{\mathrm{total}} &= |\Phi\rangle_{\mathrm{pol}} \otimes |\Phi\rangle_{\mathrm{e-t}} \\
&= \frac{1}{2}\left(|H\rangle_A |H\rangle_B + |V\rangle_A |V\rangle_B\right) \\
&\quad \otimes \left(|t\rangle_A |t\rangle_B + |t+\tau\rangle_A |t+\tau\rangle_B\right)
\end{aligned}
\tag{1}
$$

where H and V represent horizontally and vertically polarized photon states, whereas $t$ and $t+\tau$ denote photon-pair emission times with a delay $\tau$ with $t_p \gg \tau > t_c$.

Photon A was guided to a local measurement module and photon B was guided to a transmitter telescope on the roof of the institute via a 15-m-long single-mode fibre. The photons emanating from the transmitter telescope were made to overlap with a 532-nm beacon laser for pointing, acquisition and tracking and sent to a receiver telescope at BOKU via a 1.2-km-long optical free-space link. The receiver telescope consisted of a telephoto objective (Nikkor $f = 400\,\mathrm{mm}$ $f/2.8$) and an additional collimation lens. Note that the same type of objective is currently installed in the ISS Cupola module, and was recently proposed as a receiver in a quantum uplink scenario[4]. The beacon laser was transmitted through a dichroic mirror and focused onto a CCD (charge-coupled device) image sensor while the collimated single-photon beam was guided to Bob's measurement module.

The measurement modules for Alice and Bob each featured a polarization analyser and an optional transfer set-up that coupled the energy–time DOF to the polarization DOF (see also Supplementary Fig. 2). Alice's polarization analyser consisted of a variable phase shifter, a half-wave plate and a polarizing beam splitter (PBS) with multi-mode fibre-coupled single-photon avalanche diodes (SPAD) in each of its two output ports. A variable phase shift $\phi(\theta)$ could be introduced between the computational basis states $|H/V\rangle$ by tilting a birefringent YVO$_4$ crystal by an angle $\theta$ about its optical axis using a stepper motor. With the half-wave plate set to 22.5°, this configuration corresponds to a polarization measurement in a superposition

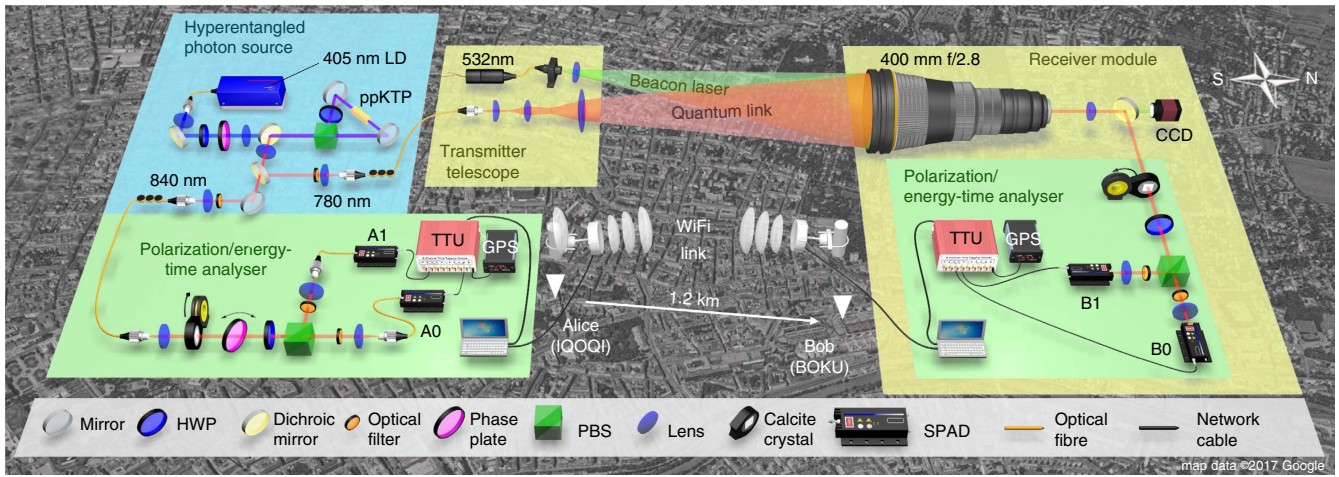

**Figure 1 | Illustration of the high-dimensional entanglement distribution experiment.** A hyperentangled photon source was located in a laboratory at the IQOQI Vienna. The source utilized SPDC in a periodically poled KTiOPO$_4$ (ppKTP) crystal, which was placed at the centre of a Sagnac interferometer and pumped with a continuous-wave 405-nm laser diode (LD). The polarization/energy–time hyperentangled photon pairs had centre wavelengths of $\lambda_B \sim 780$ nm and $\lambda_A \sim 840$ nm, respectively. Photon A was sent to Alice at IQOQI using a short fibre link, while photon B was guided to a transmitter telescope on the roof of the institute and sent to Bob at the BOKU via a 1.2-km-long free-space link. At Bob, the photons were collected using a large-aperture telephoto objective with a focal length of 400 mm. The 532-nm beacon laser was separated from the hyperentangled photons using a dichroic mirror and focused onto a CCD image sensor to maintain link alignment and to monitor atmospheric turbulence. Alice's and Bob's analyser modules allowed for measurements in the polarization or energy–time basis. The polarization was analysed using a half-wave plate (HWP) and a polarizing beam splitter (PBS) with a single-photon avalanche diode (SPAD) in each output port. An additional phase shift could be introduced in Alices measurement module by tilting a birefringent crystal about its optical axis. In both analyser modules, optional calcite crystals could be added before the PBS to introduce the polarization-dependent delay required for Franson interference measurements in the energy–time basis. Single-photon detection events were recorded with a GPS-disciplined time tagging unit (TTU) and stored on local hard drives for post processing. Bob's measurement data were streamed to Alice via a classical WiFi link to identify photon pairs in real time. Map data ⓒ2017 Google.

basis $+\phi/-\phi$, where $|\pm\phi\rangle = \frac{1}{\sqrt{2}}\left(|H\rangle \pm e^{i\phi}|V\rangle\right)$. Bob's polarization analyser module used a motorized half-wave plate and a PBS with a SPAD (active area of 180 μm) in each of its two output ports. To reduce background counts from the city, long-pass filters and interference filters were added and the optical system was engineered such that the detectors had a small field of view (225 μrad). Bob's analysis set-up allowed for measurements in any linear polarization basis, in particular the basis $+45°/-45°$, where $|\pm 45°\rangle = \frac{1}{\sqrt{2}}(|H\rangle \pm |V\rangle)$.

For the analysis of energy–time entanglement, we employed a variant of the original Franson interferometer[50] that uses polarization-dependent delays to map an energy–time subspace spanned by early $|t\rangle$ and late $|t+\tau\rangle$ pair emissions to the polarization state space[51]. This variant has the advantage that the polarization entanglement acts as a pair of synchronized switches, such that there is no need for detection post selection[54]. These unbalanced polarization interferometers at Alice and Bob were implemented with 3-mm-long calcite crystals, which could be inserted before the polarization analysers. The calcite crystal introduced a birefringent delay of $\tau \sim 2$ ps, which exceeded the coherence time of the SPDC photons but was significantly shorter than the coherence time of the pump laser. Note that the particular choice of delay restricts our considerations to a two-dimensional subspace of the intrinsically continuous-variable energy–time space. Hence, after introducing the polarization-dependent delay, polarization measurements in a superposition basis correspond to measurements of energy–time superpositions of the form $|t\rangle + e^{i\phi}|t+\tau\rangle$. A more detailed discussion is provided in the Supplementary Discussion.

The arrival times of single-photon detection events at Alice and Bob were recorded relative to local 10 MHz global positioning system (GPS)-disciplined clocks and stored on local hard drives for post processing of two-photon detection events. Bob's measurement data were also streamed to Alice via a 5 GHz directional WiFi antenna, where all combinations of two-photon detection events within a coincidence window of 2 ns were monitored on-the-fly, while compensating for relative clock drifts (see Fig. 2)[55].

**Link performance.** Directly at the source, we detected a total coincidence rate of $R^{(2)} \sim 84$ kcps and singles rates of $R_A^{(1)} \sim 400$ kcps and $R_B^{(1)} \sim 350$ kcps. Of the single photons sent via the free-space link, we measured an average of 100 kcps in Bob's two detector channels, and an average rate of $\sim 20$ kcps two-photon detection events per second. For night-time operation, the background counts were $R_{B,1}^{(1)} \sim 450-800$ cps and $R_{B,0}^{(1)} \sim 250-400$ cps for Bob's two detector channels, whereby 200 cps and 50 cps were due to intrinsic detector dark counts.

Because of atmospheric turbulence, the link transmission varied on the timescale of ms (see Fig. 2). The time-averaged beam diameter at the receiver was of the same order as the receiver aperture (14.5 cm). We observed an average total link transmission of $\sim 18\%$, including all optical losses from source to receiver, where approximately half of the transmission loss was due to absorption in optical components.

Besides being used for pointing, acquisition and tracking, the CCD image sensor also monitored angle of arrival fluctuations caused by atmospheric turbulence[56]. The full-width at half-maximum of the angular variation at the telescope was estimated with a series of short exposure images and was in the order of $\sim 25$ μrad, which corresponds to an atmospheric Fried parameter of $\sim 2$ cm at 532 mm. This is similar to that experienced in a free-space link over 144 km on the Canary islands[1] and represents a worst case scenario in a satellite communication experiment through the atmosphere. Note that the angle of arrival fluctuations were significantly smaller than the detector's field

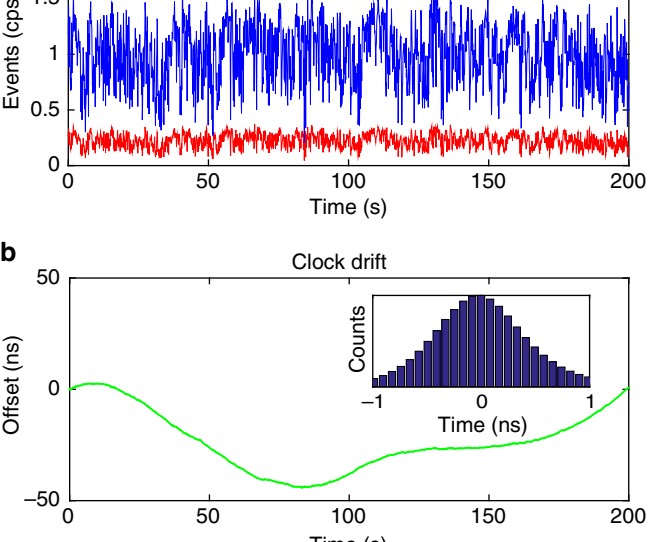

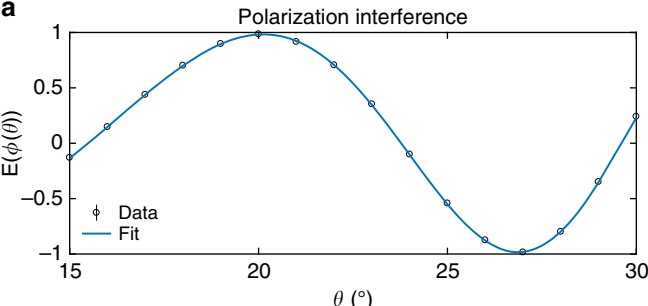

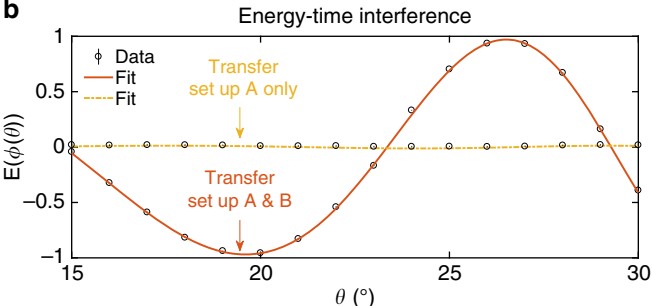

**Figure 2 | Transmission rate and clock drift.** (**a**) Average single-photon (blue line) and two-photon (red line) detection rate (100 ms integration time) after 1.2-km-long free-space transmission. The short-term signal fluctuated due to atmospheric turbulence, whereas the time-averaged rate of ~20 kcps remained almost constant over several hours. (**b**) Relative clock drift between Alice and Bob. The inset depicts the normalized histogram of two-photon detection events in 80 ps time bins centred around the flight-time offset of ~3.94 μs. All data acquired for night-time operation on 25–26 April 2016.

**Figure 3 | Experimental characterization of hyperentanglement.** Two-photon correlation functions in the polarization basis (**a**) and energy–time basis (**b**) as a function of the variable phase shift introduced in Alice's measurement module. Each data point was evaluated from two-photon detection events accumulated over a 10 s integration time, without subtraction of accidental counts. The error bars that denote the 3-σ s.d. due to Poissonian count statistics are smaller than the data markers. The best fit functions (least-mean-square fit to the expected two-photon correlation in presence of experimental imperfections) exhibit visibilities $V_{\text{pol}}^{\text{fit}} = 98.3 \pm 0.5\%$ in the polarization basis (blue line) and $V_{\text{e-t}}^{\text{fit}} = 96.8 \pm 1\%$ in the energy–time basis (orange line). Almost no interference was observed when the energy–time to polarization transfer set-up was introduced in Alice's detection module only (yellow line, $V_{\text{e-t}}^{\text{fit}} = 1 \pm 1\%$).

of view; the fluctuation of detected count rates visible in Fig. 2 stem from beam wander at the aperture of the receiver telescope.

**Experimental visibilities.** To verify the integrity of the atmospheric quantum communication channel for hyperentangled photons, we first assessed experimental two-photon polarization correlation:

$$E = \frac{N_{1,1} + N_{0,0} - N_{1,0} - N_{0,1}}{\sum_{ij} N_{i,j}} \quad (2)$$

where $N_{i,j}$ denotes the number of coincidence counts between Alice and Bob's SPAD detectors ($i, j \in \{1, 0\}$). We define the experimental visibility $V$ in the superposition basis as the maximum correlation observed while scanning the phase of Alice's measurement basis $+\phi / -\phi$ and keeping Bob's measurement set-up in the linear $+45° / -45°$ polarization basis, that is, $V = \max_\theta(|E(\phi(\theta))|)$. The scan over the polarization correlations depicted in Fig. 3 exhibited a maximum value of $V_{\text{pol}}^\phi = 98.5 \pm 0.15\%$. The correlation in the linear H/V measurement basis was $V_{\text{pol}}^{\text{H/V}} = 99.33 \pm 0.015\%$. Note that the H/V visibility is limited almost exclusively due to accidental coincidences.

Similarly, with the transfer set-up inserted in both measurement modules, we observed Franson interference with a visibility of $V_{\text{e-t}}^\phi = 95.6 \pm 0.3\%$ (Fig. 3). To verify that the high visibility is due to two-photon energy–time entanglement, and not single-photon interference of photons A and B independently, we removed the transfer set-up in Bob's detection module. In this case, the measurement outcomes were completely uncorrelated, irrespective of $\phi(\theta)$, since the polarization-

dependent delay exceeded the coherence time of the SPDC photons. This is indicated by the straight line in Fig. 3.

**Lower bounds on entanglement.** The experimental visibilities establish lower bounds of $0.978 \pm 0.0015$ and $0.912 \pm 0.006$ on the concurrence[57] in the polarization space and energy–time subspace, respectively (see 'Methods' section). These values correspond to respective minimum values of $0.940 \pm 0.004$ and $0.776 \pm 0.014$ ebits of entanglement of formation.

In the 'Methods' section, we use these values to establish a lower bound for the Bell-state fidelity $\mathcal{F}(\hat{\rho}_{\text{pol,e-t}})$ of the hyperentangled state of the combined system (see also Supplementary Discussion). We achieve this by formulating this lower bound as a semidefinite programming (SDP) problem, in which we minimize the four-dimensional concurrence and fidelity to a four-dimensional Bell state over all possible states in the combined Hilbert space that satisfy the experimentally observed subspace concurrences. We obtain lower bounds of 1.4671 ebits of entanglement of formation and a Bell-state fidelity of 0.9419, thus certifying four-dimensional entanglement[58].

**Discussion**

We have distributed hyperentangled photon pairs via an intra-city free-space link under conditions of strong atmospheric turbulence. Despite the severe wavefront distortions, we observed a high two-photon detection rate of ~20 kcps over a link distance

of 1.2 km. In a series of experiments, we independently observed high-visibility two-photon interference in the two-dimensional polarization state space and a two-dimensional energy–time subspace. These visibilities are sufficient to certify entanglement in both subspaces individually, and, for the first time, the coherent transmission of genuine high-dimensional quantum entanglement via a real-world free-space link. While the transmission of polarization-entangled photons has been studied in a number of previous field trials, our results now demonstrate the feasibility of exploiting energy–time/polarization hyperentanglement in real-world link conditions with strong atmospheric turbulence.

Our analysis of interference in the energy–time DOF relies on an unbalanced polarization interferometer that coherently couples the polarization space with a two-dimensional energy–time subspace. The current approach of mapping the time-bin entanglement to the polarization DOF is of course intrinsically limited to accessing two-dimensional subspaces of the high-dimensional energy–time space. As recent experiments have clearly shown[59,60], the potential dimensionality of energy–time entanglement is orders of magnitudes larger. In fact, theoretically, it should only be limited by the achievable number of time bins within the coherence time of the pump laser. The main challenge remains the implementation of superposition measurements, where a single calcite is inherently limited to two dimensions. Future set-ups for free-space experiments could use several delay lines, or a variable delay line[25], to greatly increase the dimensionality and with it the resistance to inevitable background noise.

Critically, the energy–time to polarization transfer set-up can be understood as an implementation of a single-photon two-qubit operation[37], which can be exploited in, for example, hyperentanglement-assisted Bell-state measurements and efficient entanglement purification schemes[47–49,61]. To fully benefit from hyperentanglement in such applications, the delay between early and late photon arrival times will have to be directly resolved by the detectors. The main challenge therein lies in maintaining a constant phase relation between the long and short arms of the unbalanced interferometers for distorted input beams with a wide range of angles of incidence. However, such free-space compatible time-bin analysers have recently been demonstrated in refs 34,35, where the issue was ingeniously tackled via the implementation of a 4-$f$ imaging system in the long arm of the interferometer.

The coherent transmission of quantum information embedded in a genuine high-dimensional state space under real-world link conditions represents an important step towards long-distance quantum communications with more complex quantum systems and could play a key role in the implementation of advanced quantum information processing protocols in the future. A large quantum state space not only allows for larger information capacity in quantum communication links, as well as devising quantum communication schemes with more resilience against noise and improved security against eavesdroppers, but also allows for more diverse types of non-classical correlations, which could prove vital in addressing technological challenges on the path towards global-scale quantum networks, as well as fundamental physics experiments.

Since polarization-entangled photon sources based on SPDC quite naturally exhibit energy–time entanglement when pumped with a continuous-wave pump laser, the approach can readily be implemented with existing sources and proposals for satellite-link experiments with polarization-entangled photons without need for additional critical hardware[4,62–64]. The additional possibility of analysing energy–time entanglement could provide a platform for entirely new fundamental physics experiments with long-distance satellite links, such as the evaluation of models for gravity-induced wave function collapse[65] or quantum information processing in a

relativistic framework[66]. High-dimensional energy–time entangled states can also be considered as a natural candidate for applications in quantum-enhanced clock synchronization protocols[67], and could allow for significant gains in performance by employing other quantum features, such as non-local cancellation of dispersion[68]. We also believe that our results will motivate both further theoretical research into energy–time entanglement experiments conceivable at relativistic scenarios with satellite links, as well as experimental research into the exploitation of hyperentanglement in long-distance quantum communications.

## Methods

**Hyperentangled photon source.** The hyperentangled photon source was based on type-0 SPDC in a periodically poled KTiOPO$_4$ (ppKTP) crystal. The ppKTP crystal was bidirectionally pumped inside a polarization Sagnac interferometer[52,53] and generated polarization-entangled photon pairs with centre wavelengths $\lambda_A \sim 840$ nm and $\lambda_B \sim 780$ nm. Photons A and B were separated using a dichroic mirror and coupled into optical single-mode fibres. For a pump power of 400 μW incident on the crystal, we detected a pair rate of of $R^{(2)} \sim 84$ kcps and singles rates of $R_A^{(1)} \sim 400$ kcps and $R_B^{(1)} \sim 350$ kcps directly after the source's single-mode fibres. This corresponds to a normalized detected pair rate of 200 kcps mW$^{-1}$ and a detected spectral brightness of 100 kcps mW$^{-1}$ nm$^{-1}$. Without correcting for background counts, losses or detection inefficiency, we measure an average coincidence-to-singles ratio $R^{(2)}/\sqrt{R_B^{(1)} R_A^{(1)}} \sim 0.22$.

The quasi-phase matching condition in the 20-mm-long ppKTP crystal[69] resulted in a spectral bandwidth of $\Delta\lambda \sim 2$ nm, which corresponds to a two-photon coherence time of $t_c \lesssim 1$ ps. The emission time of a photon pair is uncertain within the significantly longer coherence time of the continuous-wave grating stabilized pump laser diode ($t_p \gtrsim 100$ ns, such that the bi-photon state is in a superposition of possible pair-emission times (see also Supplementary Fig. 1), that is, entangled in the energy–time DOF[50].

**Energy–time visibility measurement.** We employed a variant of the original Franson scheme[50,51] with unbalanced polarization interferometers to assess the coherence of the energy–time entanglement. The polarization interferometers were implemented with birefringent calcite crystals, which introduced a polarization-dependent time shift $\tau$ (Fig. 4). The particular choice of delay defines a two-dimensional subspace (of the intrinsically continuous-variable energy–time space) spanned by the time-delayed basis states $|t\rangle$ and $|t+\tau\rangle$. Since this delay is significantly shorter than the timing resolution of the detectors, our experimental results can be understood as averages over a larger state space in the energy–time domain. The maximally entangled Bell state in this subspace reads:

$$|\Phi\rangle_{e-t} = \frac{1}{\sqrt{2}} \left( |t\rangle_A |t\rangle_B + |t+\tau\rangle_A |t+\tau\rangle_B \right) \quad (3)$$

In the Supplementary Discussion (see also ref. 54), we show how the transfer set-up in combination with polarization entanglement is used to probe the experimental density matrix $\rho'_{e-t}$ in the energy–time subspace. After introducing a polarization-dependent time shift for Alice and Bob, the visibility of polarization measurements in the superposition basis is determined by the off-diagonal coherence terms via:

$$V_{e-t}^{\phi} \sim \left| \langle t, t | \rho'_{e-t} | t+\tau, t+\tau \rangle \right| \quad (4)$$

The total state space accessed in our experiment thus comprises the two-dimensional polarization space and an effectively two-dimensional energy–time subspace. The hyperentangled state of the total system can be expressed as a

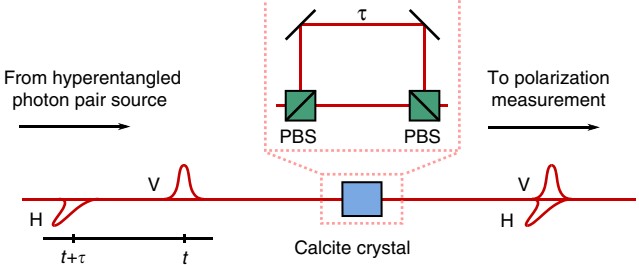

**Figure 4 | Energy–time to polarization transfer set-up.** The calcite crystal acts as an unbalanced polarization interferometer, which introduces a time shift of $\tau > t_c$ between vertically (V) and horizontally (H) polarized photons. After the transfer set-up polarization measurements in a superposition basis allow to probe energy–time coherence (see also Supplementary Discussion).

maximally entangled state in four dimensions:

$$|\Psi\rangle_{\text{total}} = \frac{1}{2}\left(|0\rangle_A|0\rangle_B + |1\rangle_A|1\rangle_B + |2\rangle_A|2\rangle_B + |3\rangle_A|3\rangle_B\right) \qquad (5)$$

with basis vectors $|0\rangle = |H,t\rangle$, $|1\rangle = |H,t+\tau\rangle$, $|2\rangle = |V,t\rangle$, $|3\rangle = |V,t+\tau\rangle$. For more details, refer to the the Supplementary Discussion.

**Certification of entanglement.** In ref. 57, easily computable lower bounds for the concurrence of mixed states that have an experimental implementation were derived:

$$\mathcal{C}(\rho) \geq 2\times\text{Re}(\langle00|\rho|11\rangle) - (\langle01|\rho|01\rangle + \langle10|\rho|10\rangle) \qquad (6)$$

where $\rho$ is the density matrix in the two-dimensional subspace. In the Supplementary Discussion, we show the concurrence can be related to the experimental polarization space and energy–time visibilities via:

$$\begin{aligned}\mathcal{C}(\rho_{\text{pol}}) &\geq V_{\text{pol}}^{\phi} + V_{\text{pol}}^{\text{H/V}} - 1 \\ \mathcal{C}(\rho_{\text{e}-\text{t}}) &\geq 2\times V_{\text{e}-\text{t}}^{\phi} - 1\end{aligned} \qquad (7)$$

Note that the bound on the energy–time concurrence involves the additional assumption that there is no phase relationship between accidental coincidence that occur in time bins separated by more than the coherence time. We believe that, while this assumption precludes a certification of entanglement that meets the requirements for quantum cryptography, it is completely justified for our proof-of-concept experiment. This also agrees with our experimental observation that scanning the phase of the entangled state in the source had no effect on the single-photon coherence.

With the experimentally obtained lower bounds for $\mathcal{C}(\rho_{\text{pol}})$ and $\mathcal{C}(\rho_{\text{e}-\text{t}})$ at hand, we calculate a lower bound for the concurrence of the global state $\mathcal{C}(\rho_{\text{pol},\text{e}-\text{t}})$ by solving the following convex optimization problem: a minimization of the function that defines a lower bound for the concurrence, over all states $\rho$ acting on a four-dimensional Hilbert space such that the concurrence of the reduced states in two-dimensional subspaces satisfy the constraints of being lower bounded by the values $\mathcal{C}(\rho_{\text{pol}})$ and $\mathcal{C}(\rho_{\text{e}-\text{t}})$. As demonstrated in the Supplementary Discussion, this convex optimization problem has a SDP characterization and satisfies the condition of strong duality. Hence, the obtained lower bound of $\mathcal{C}(\rho_{\text{pol},\text{e}-\text{t}}) \geq 1.1299$ has an analytical character.

Another useful measure of entanglement is the entanglement of formation $E_{\text{oF}}(\rho)$, which represents the minimal number of maximally entangled bits (ebits) required to produce $\rho$ via an arbitrary local operations and classical communication procedure. It can be shown[70] that the entanglement of formation is lower bounded by the concurrence according to:

$$E_{\text{oF}}(\rho) \geq -\log\left(1 - \frac{\mathcal{C}(\rho)^2}{2}\right) \qquad (8)$$

Hence, from the lower bound for the concurrence $\mathcal{C}(\rho_{\text{pol},\text{e}-\text{t}})$, it is possible to calculate a lower bound of $E_{\text{oF}}(\rho_{\text{pol},\text{e}-\text{t}}) \geq 1.4671$ for the entanglement of formation, which is sufficient to certify three-dimensional bipartite entanglement[70].

By adapting the objective function of our SDP from the concurrence to the fidelity to the maximally entangled four-dimensional state, it is possible to lower bound the latter quantity by performing a minimization over the same variable and same constraints. As shown in the Supplementary Discussion, this second SDP also satisfies strong duality and provides the analytical bound of $\mathcal{F}(\rho_{\text{pol},\text{e}-\text{t}}) \geq 0.9419$, which certifies four-dimensional bipartite entanglement[58].

**Data availability.** Data supporting our experimental findings are available from the corresponding authors on reasonable request.

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

## Acknowledgements

We thank Johannes Handsteiner, Dominik Rauch and Sören Wengerowsky for their support in setting up the experiment. We also thank Mario Krenn and Sven Ramelow for helpful conversations and comments on the initial draft of the manuscript. We also thank the Bundesimmobiliengesellschaft (BIG) for providing the room for our receiving station. Financial support from the Austrian Research Promotion Agency (FFG)—Agentur für Luft—und Raumfahrt (FFG-ALR contract 844360), the European Space Agency (ESA contract 4000112591/14/NL/US), the Austrian Science Fund (FWF) through (P24621-N27) and the START project (Y879-N27), as well as the Austrian Academy of Sciences is gratefully acknowledged.

## Author contributions

F.S. conceived the experiment. S.E. designed and developed the entangled photon source and local detection module under the supervision of F.S. M.F. designed and developed the free-space link and receiver optics with help from T.S. F.S., S.E. and M.F. performed the experiment under the guidance of R.U. M.F. and B.L. designed the coincidence tracking software and processed detection events. F.S., M.F. and S.E. analysed the experimental data. F.S., M.F., T.S., S.E. and R.U. discussed and evaluated the experimental results. M.H. and J.B. established the bounds on high-dimensional entanglement and provided theory support. F.S. wrote a first draft and all authors contributed to the final version of the manuscript.

## Additional information

**Competing interests:** The authors declare no competing financial interests.

