## [Peer Review File · Nature Communications]

Reviewer #1 (Remarks to the Author):

The manuscript by Steinlechner et al. reports on an experimental distribution of hyper-entanglement over a 1.2 km free-space link. The authors utilize the photons from parametric down conversion to demonstrate hyper-entanglement of spatially separated photons in polarization and time-energy bases. The manuscript provides a well written survey of the earlier work on this topic, the theoretical concepts, and the experimental results. However, the paper might be inappropriate for a publication in Nature Communication because it's lack of breakthrough either in concept or technological achievements. Considering this, I recommend the publication of the paper in a more specialized journal.

Below, I have listed a few comments regarding this recommendation:

1. Free-space quantum communication has been a focus of several recent publications (fully cited in the manuscript). The current work does not provide any technological improvement over the previous work in terms of key figures of merits such as the operation range, bit rate, adaptive optics, etc. Further, the paper does not contribute to any new theoretical methods, or principles of operation (Franson interferometry is a well-established concept in the community).
2. The major contribution of the current work is the implementation of a time-energy entanglement distribution over a long-range free-space link. This could be considered a significant achievement if the system reached its essential goal of distributing high-dimensional entanglement. The current implementation, however, is only limited to a two-dimensional (time-energy) sub-space. This is due to the inherent difficulty of performing interferometry with the scrambled mode profile of a beam that has propagated through atmospheric turbulence. This is a rather fundamental issue that has hindered the realization of a true high-dimensional protocol over free-space links thus far, and merits a more detailed discussion by the authors (They have briefly mentioned this point at the end of the manuscript).
3. It would be useful to the community if the authors provide an estimate of the performance of their system for a operation in a large dimensional Hilbert space. Such an analysis should include the future steps required to achieve the full potential of the system. Also, I suggest the authors use the relevant coherence times to estimate the Schmidt number of the entangled biphoton states in their experiment.

4. The authors have provided the $f\#$ of their telescope, but I could not find the diameter of the beam in the experiment. It would be useful if they can compare the beam size and the telescope diameter, against the average beam displacements caused by turbulence (they currently report the average angular deviation, but the actual displacement can be more useful for some readers). Also, it would be beneficial if the authors comment on possible enhancements that can be realized using adaptive optics.

Reviewer #2 (Remarks to the Author):

The authors describe a study about free space quantum communication using hyper-entangled states. This is certainly a result at the forefront of the field. I find the manuscript is very well written, the results strongly support their claim, and the results are very impressive. I am supportive of moving forward to have this published in Nature Communications.

For someone outside of the immediate field, that nature of the states might be difficult to understand. I encourage the authors to consider adding a figure (or figure inset) that illustrates the nature of the quantum states.

Reviewer #3 (Remarks to the Author):

The manuscript by Steinlechner et al. details the distribution of hyper-entanglement via an intra-city freespace link. This is a very interesting topic, and I think that the paper is suitable for the general science audience of Nature Communications. The quality of the submission is very high, with high quality results and high quality presentation.

I think that there are two recent papers that the authors should cite. I have pasted the links to these below:

<http://www.nature.com/nphoton/journal/v10/n10/full/nphoton.2016.179.html>

<http://www.nature.com/nphoton/journal/v10/n10/full/nphoton.2016.180.html>

It would be good to put the current submission in the context of this prior art.

My main comment is about quantifying the results. The authors wrote about the importance of “high-fidelity” transmission, but they do not state their fidelity of their transmitted state. I think that this is something that would make the paper stronger if it was included.

Overall, I think that the paper is suitable for publication with very minor changes to the work.

Point-by-point response:

Reviewer #1:

The manuscript by Steinlechner et al. reports on an experimental distribution of hyper-entanglement over a 1.2 km free-space link. The authors utilize the photons from parametric down conversion to demonstrate hyper-entanglement of spatially separated photons in polarization and time-energy bases. The manuscript provides a well written survey of the earlier work on this topic, the theoretical concepts, and the experimental results.

We thank the reviewer for the positive feedback on the overall quality of our work. The reviewer also expresses doubts regarding novelty:

However, the paper might be inappropriate for a publication in Nature Communication because it's lack of breakthrough either in concept or technological achievements. Considering this, I recommend the publication of the paper in a more specialized journal.

Below, I have listed a few comments regarding this recommendation:

1. Free-space quantum communication has been a focus of several recent publications (fully cited in the manuscript). The current work does not provide any technological improvement over the previous work in terms of key figures of merits such as the operation range, bit rate, adaptive optics, etc. Further, the paper does not contribute to any new theoretical methods, or principles of operation (Franson interferometry is a well-established concept in the community).
2. The major contribution of the current work is the implementation of a time-energy entanglement distribution over a long-range free-space link. This could be considered a significant achievement if the system reached its essential goal of distributing high-dimensional entanglement. The current implementation, however, is only limited to a two-dimensional (time-energy) sub-space. . . .

We agree with the reviewer, that Franson interferometry is a well-established concept in the community. Nevertheless, the extension of this established concept to new operational regimes, in our case long-distance free-space communications, has resulted in several recent publications [9, 17]. After over a decade of research in free-space quantum communications, our work is the first to demonstrate time-energy entanglement after propagation through a turbulent free-space link. This is particularly relevant for quantum experiments with satellites, where space-proof polarization entangled photon sources are at the brink of being launched into space. Our results show that the exact same sources in space can in principle also be used to measure time-energy entanglement. This could significantly extend the scope of future experiments beyond mere Bell-test experiments and quantum key distribution in space.

While we believe this could constitute a major novelty in its own right, we must point out that this was not the sole novelty of our work. We have also shown hyperentanglement in energy-time *and* polarization to realize an easily accessible extension to the state space for ongoing and future free-space experiments. In the methods section we point this out more clearly now:

”The total state space accessed in our experiment thus comprises the 2-dimensional polarization space and an effectively 2-dimensional energy-time subspace. The hyper-entangled state of the total system can be expressed as a maximally-entangled state in four dimensions:

$$|\Psi\rangle_{\text{total}} = \frac{1}{2} (|0\rangle_A|0\rangle_B + |1\rangle_A|1\rangle_B + |2\rangle_A|2\rangle_B + |3\rangle_A|3\rangle_B) \quad (1)$$

with basis vectors $|0\rangle = |H, t\rangle$, $|1\rangle = |H, t + \tau\rangle$, $|2\rangle = |V, t\rangle$, $|3\rangle = |V, t + \tau\rangle$. For more details refer to the the supplementary material.”

We have made a number of additions in order to argue this aspect of our work more convincingly; We show that, using hyperentanglement, we were able to distribute high-dimensional entanglement via a free-space link for the first time. We have made changes throughout the manuscript to more clearly distinguish the scope of our work from prior state-of-the-art. In the methods section and the theory supplement, we show that our measurement results establish a lower bound for the Bell-state fidelity of the hyper-entangled state of the combined system. We obtain lower bounds of 1.4671 ebits of entanglement of formation and a Bell state fidelity of 0.9419, which is sufficient to certify 4-dimensional entanglement. We thus believe that this major point of concern has been addressed in the revised manuscript.

2. *continued...* This is due to the inherent difficulty of performing interferometry with the scrambled mode profile of a beam that has propagated through atmospheric turbulence. This is a rather fundamental issue that has hindered the realization of a true high-dimensional protocol over free-space links thus far, and merits a more detailed discussion by the authors (They have briefly mentioned this point at the end of the manuscript).

Our experimental results are direct proof that interferometry can be performed despite the scrambled mode profile. Two other groups (Refs.[5, 16]) have also recently demonstrated unbalanced interferometers that maintain a fixed phase for distorted mode profiles and a wide range of angles-of-incidence. This was achieved by adding a 4-f imaging system to the longer arm of the unbalanced interferometer and imaging the distorted input wavefront to the output beam splitter. This effectively solves issues with the scrambled wavefront and angle-of-arrival jitter in free-space links. We have also added a clarifying statement in the discussion and refer to the excellent articles [5, 16] for a detailed description of the principle functionality:

”Critically, our 2-dimensional transfer setup can also be understood as an implementation of single-photon two-qubit operations [2], which can be exploited in e.g. hyperentanglement-assisted Bell state measurements and efficient entanglement purification schemes [10, 13, 12, 11]. In order to fully benefit from hyperentanglement in such applications, the delay between early and late photon arrival times will have to be directly resolved by the detectors. The main challenge therein lies in maintaining a constant phase relation between the long and short arms of the unbalanced interferometers for distorted input beams with a wide range of angles-of-incidence. However, such free-space compatible time-bin analyzers have recently been demonstrated [5, 16], where the issue was ingeniously tackled via the implementation of a 4- f imaging system in the long arm of the interferometer.”

3. It would be useful to the community if the authors provide an estimate of the performance of their system for a operation in a large dimensional Hilbert space. Such an analysis should include the future steps required to achieve the full potential of the system. Also, I suggest the authors use the relevant coherence times to estimate the Schmidt number of the entangled biphoton states in their experiment.

We agree that this would be interesting and have added a discussion of the high-dimensional nature of the entangled states in the supplementary material. Furthermore, we have added a paragraph elaborating possible next steps to the discussion:

”Our analysis of interference in the energy-time DOF relies on an unbalanced polarization interferometer that coherently couples the polarization space with a 2-dimensional energy-time subspace. The current approach of mapping the time-bin entanglement to the polarization degree of freedom is of course intrinsically limited to accessing two-dimensional subspaces of the high dimensional energy time space. As recent experiments have clearly shown [15, 8], the potential dimensionality of energy-time entanglement is orders of magnitudes larger. In fact, theoretically, it should only be limited by the achievable time-bin vs the coherence time of the laser. The main challenge remains the implementation of superposition measurements, where a single calcite is inherently limited to two dimensions. Future setups for free-space experiments could use several delay lines, or a variable delay line [17] to greatly increase dimensionality and with it the resistance to inevitable background noise.”

At this point we would like to point out that there are numerous intriguing experiments that would benefit already from the four-dimensional Hilbert Space that can be accessed using a single delay line (e.g. hyper-entanglement-assisted Bell-state measurements), as elaborated in the discussion section of the manuscript.

4. The authors have provided the f -number of their telescope, but I could not find the diameter of the beam in the experiment. It would be useful if they can compare the beam size and the telescope diameter, against the average

beam displacements caused by turbulence (they currently report the average angular deviation, but the actual displacement can be more useful for some readers)...

We thank the reviewer for pointing this out. We agree that this would be of interest for readers from the field of free-space optical communications, and have added the f-number of the telescope. The section now reads:

”Due to atmospheric turbulence the link transmission varied on the time-scale of ms (see Fig. 2). The time-averaged beam diameter at the receiver was of the same order as the receiver aperture (14.5cm).”

...Also, it would be beneficial if the authors comment on possible enhancements that can be realized using adaptive optics.

Adaptive optics would be an interesting addition that could increase the transmission rate in future experiments, in particular for longer link distances. However, we don't believe that this benefit is necessarily specific to energy-time entanglement or hyper-entanglement in polarization and energy-time. In this respect it seems that the adaptive optics are not fundamentally required, as indicated by the high Franson visibility we observed despite severe atmospheric turbulence and the recently demonstrated free-space compatible time-bin analyzers in Refs.[5, 16]. As mentioned in the introduction, the benefits of using active wavefront correction will be more significant for other degrees of freedom (in particular spatial mode encoding).

In conclusion, we thank the reviewer for numerous helpful comments relating to various technical aspects of the article. We are convinced that the reviewers concerns regarding novelty have been addressed and believe that the reviewers suggestions have contributed to a significantly improved revised manuscript.

Reviewer #2:

The authors describe a study about free space quantum communication using hyper-entangled states. This is certainly a result at the forefront of the field. I find the manuscript is very well written, the results strongly support their claim, and the results are very impressive. I am supportive of moving forward to have this published in Nature Communications.

We thank the reviewer for this positive assessment and are glad to read that the reviewer is in favour of publishing our work in Nature Communications.

For someone outside of the immediate field, that nature of the states might be difficult to understand. I encourage the authors to consider adding a figure (or figure inset) that illustrates the nature of the quantum states.

We agree that the nature of the time-energy entangled states, as well as their measurement using a transfer setup might be hard to grasp for readers outside the immediate field. In response, we now write the energy-time state only for the two-dimensional sub-space, as we believe that this should be clearer:

”In our proof of concept demonstration we focused on a two-dimensional subspace of the high-dimensional energy-time space (see methods). The total state space considered in our proof-of-concept experiment is thus a 4-dimensional hyperentangled state in polarization and energy-time:

$$|\Psi\rangle_{\text{total}} = |\Phi\rangle_{\text{pol}} \otimes |\Phi\rangle_{\text{e-t}} = \frac{1}{2} (|H\rangle_A |H\rangle_B + |V\rangle_A |V\rangle_B) \otimes (|t\rangle_A |t\rangle_B + |t + \tau\rangle_A |t + \tau\rangle_B) \quad (2)$$

where H and V represent horizontally and vertically polarized photon states whereas t and $t + \tau$ denote photon-pair emission times with a delay τ with $t_p \gg \tau > t_c$. The subscripts A and B label the respective single-mode fiber for Alice and Bob.”

We have added a figure (Fig. 4) illustrating the transfer of time-energy states to polarization states in the methods section. We hope this will allow the non-specialist readers to get an immediate intuitive understanding of the measurement setup and the nature of the quantum states in our experiment. In the methods we now write:

”Energy-time visibility measurement

We employed a variant of the original Franson scheme [14, 3] with unbalanced polarization interferometers to assess the coherence of the energy-time state. The polarization interferometers were implemented with birefringent calcite crystals, which introduced a polarization-dependent time shift τ (Fig. 4). The particular choice of delay defines a 2-dimensional subspace (of the intrinsically continuous-variable energy-time space) spanned by the time-delayed basis states $|t\rangle$ and $|t + \tau\rangle$. Since this delay is significantly shorter than the timing resolution of the detectors, our experimental results can be understood as averages over a larger state space in the energy-time domain. The maximally-entangled Bell state in this subspace reads:

$$|\Phi\rangle_{e-t} = \frac{1}{\sqrt{2}} (|t\rangle_A |t\rangle_B + |t+\tau\rangle_A |t+\tau\rangle_B) \quad (3)$$

In the Supplementary Material (see also Ref. [6]) we show how the transfer setup in combination with polarization entanglement is used to probe the experimental density matrix ρ'_{e-t} in the energy-time subspace. After introducing a polarization-dependent time shift for Alice and Bob, the visibility of polarization measurements in the superposition basis is determined by the off-diagonal coherence terms via:

$$V_{e-t}^\phi \sim |\langle t, t | \rho'_{e-t} | t + \tau, t + \tau \rangle| \quad (4)$$

”

Additionally, we have written an extensive theory supplement which explains these points in more detail. We believe that the main concepts of the manuscript can now be grasped even by non-specialists, without necessarily referring to the additional literature links in the references. We thank the reviewer for pointing this out to us, and believe that the changes made in response to this suggestion have made the article accessible to a broader readership.

Reviewer #3:

The manuscript by Steinlechner et al. details the distribution of hyper-entanglement via an intra-city freespace link. This is a very interesting topic, and I think that the paper is suitable for the general science audience of Nature Communications.

We thank the reviewer for this very positive assessment of our work and are happy that he/she considers our paper to be suitable for publication with very minor changes.

The quality of the submission is very high, with high quality results and high quality presentation.

I think that there are two recent papers that the authors should cite. I have pasted the links to these below:

<http://www.nature.com/nphoton/journal/v10/n10/full/nphoton.2016.179.html>

<http://www.nature.com/nphoton/journal/v10/n10/full/nphoton.2016.180.html>

It would be good to put the current submission in the context of this prior art.

We agree that the recent teleportation experiments are relevant to the reader, and have added the references in the introductory section.

My main comment is about quantifying the results. The authors wrote about the importance of high-fidelity transmission, but they do not state their fidelity of their transmitted state. I think that this is something that would make the paper stronger if it was included.

Overall, I think that the paper is suitable for publication with very minor changes to the work.

We fully agree that this was a shortcoming in the initial submission. We thank the reviewer for pointing this out to us and believe that the more diligent quantification of entanglement, which is now provided, has dramatically improved the manuscript. We now state lower bounds for concurrence, entanglement of formation, and the Bell-state fidelity in a new subsection of the results section:

”Lower bounds on entanglement

The experimental visibilities establish lower bounds of 0.978 ± 0.0015 and 0.912 ± 0.006 on the concurrence[7] in the polarization space and energy-time sub-space, respectively (see methods). These values correspond to respective minimum values of 0.940 ± 0.004 and 0.776 ± 0.014 ebits of entanglement of formation.

In the methods section and the theory supplement, we use these values to establish a lower bound for the Bell-state fidelity $\mathcal{F}(\hat{\rho}_{\text{pol,e-t}})$ of the hyperentangled state of the combined system. We achieve this by formulating this lower bound as a semidefinite programming problem, in which we minimize the 4-dimensional concurrence and fidelity

to a 4-dimensional Bell state over all possible states in the combined Hilbert space that satisfy the experimentally observed subspace concurrences. We obtain lower bounds of 1.4671 ebits of entanglement of formation and a Bell state fidelity of 0.9419, thus certifying 4-dimensional entanglement [1]. ”

We outline how these values were obtained in the methods section and discuss the specific details in an extensive theory supplement. The addition to the methods section reads:

”Certification of entanglement

In Ref. [7] easily computable lower bounds for the concurrence of mixed states that have an experimental implementation were derived:

$$\mathcal{C}(\rho) \geq 2 \times \text{Re}(\langle 00|\rho|11\rangle) - (\langle 01|\rho|01\rangle + \langle 10|\rho|10\rangle) \quad (5)$$

where ρ is the density matrix in the 2-dimensional subspace. In the supplementary material we show the concurrence can be related to the experimental polarization space and energy-time visibilities via:

$$\begin{aligned} \mathcal{C}(\rho_{\text{pol}}) &\geq V_{\text{pol}}^\phi + V_{\text{pol}}^{H/V} - 1 \\ \mathcal{C}(\rho_{\text{e-t}}) &\geq 2 \times V_{\text{e-t}}^\phi - 1 \end{aligned} \quad (6)$$

Note that the bound on the energy-time concurrence involves the additional assumption that there is no phase relationship between accidental coincidence that occur in time bins separated by more than the coherence time. We believe that, while this assumption precludes a certification of entanglement that meets the requirements for quantum cryptography, it is completely justified for our proof of concept experiment. This also agrees with our experimental observation that scanning the phase of the entangled state in the source had no effect on the single-photon coherence.

With the experimentally obtained lower bounds for $\mathcal{C}(\rho_{\text{pol}})$ and $\mathcal{C}(\rho_{\text{e-t}})$ at hand, we calculate a lower bound for the concurrence of the global state $\mathcal{C}(\rho_{\text{pol,e-t}})$ by solving the following convex optimization problem: a minimization of the function that defines a lower bound for the concurrence, over all states ρ acting on a 4-dimensional Hilbert space such that the concurrence of the reduced states in 2-dimensional subspaces satisfy the constraints of being lower bounded by the values $\mathcal{C}(\rho_{\text{pol}})$ and $\mathcal{C}(\rho_{\text{e-t}})$. As demonstrated in the Supplementary Material, this convex optimization problem has a semidefinite programming (SDP) characterization and satisfies the condition of strong duality. Hence, the obtained lower bound of $\mathcal{C}(\rho_{\text{pol,e-t}}) \geq 1.1299$ has an analytical character.

Another useful measure of entanglement is the entanglement of formation $E_{\text{oF}}(\rho)$, which represents the minimal number of maximally entangled bits (ebits) required to produce ρ via an arbitrary local operations and classical communication (LOCC) procedure. It can be shown [4] that the entanglement of formation is lower bounded by the

concurrence according to:

$$E_{\text{oF}}(\rho) \geq -\log \left(1 - \frac{\mathcal{C}(\rho)^2}{2} \right). \quad (7)$$

Hence, from the lower bound for the concurrence $\mathcal{C}(\rho_{\text{pol,e-t}})$ it is possible to calculate a lower bound of $E_{\text{oF}}(\rho_{\text{pol,e-t}}) \geq 1.4671$ for the entanglement of formation, which is sufficient to certify 3-dimensional bipartite entanglement [4].

By adapting the objective function of our SDP from the concurrence to the fidelity to the maximally entangled 4-dimensional state, it is possible to lower bound the latter quantity by performing a minimization over the same variable and same constraints. As shown in the Supplementary Material, this second SDP also satisfies strong duality and provides the analytical bound of $\mathcal{F}(\rho_{\text{pol,e-t}}) \geq 0.9419$, which certifies 4-dimensional bipartite entanglement [1].”

Again, we express our thanks to the reviewer for pointing out the need to better quantify the results. We believe the additions made in response to this comment have, undoubtedly and significantly, improved the overall quality of the revised manuscript.

References

- [1] Fickler, R., Lapkiewicz, R., Huber, M., Lavery, M. P., Padgett, M. J., and Zeilinger, A. (2014). Interface between path and orbital angular momentum entanglement for high-dimensional photonic quantum information. *Nature Communications*, 5:4502.
- [2] Fiorentino, M. and Wong, F. N. C. (2004). Deterministic controlled-not gate for single-photon two-qubit quantum logic. *Phys. Rev. Lett.*, 93:070502.
- [3] Franson, J. D. (1989). Bell inequality for position and time. *Physical Review Letters*, 62(19):2205–2208.
- [4] Huber, M. and de Vicente, J. I. (2013). Structure of multidimensional entanglement in multipartite systems. *Phys. Rev. Lett.*, 110:030501.
- [5] Jin, J., Agne, S., Bourgoïn, J.-P., Zhang, Y., Jennewein, T., et al. (2015). Efficient time-bin qubit analyzer compatible with multimode optical channels. *arXiv preprint arXiv:1509.07490*.
- [6] Langford, N. K. (2007). *Encoding, manipulating and measuring quantum information in optics*. PhD thesis, The University of Queensland.
- [7] Ma, Z.-H., Chen, Z.-H., Chen, J.-L., Spengler, C., Gabriel, A., and Huber, M. (2011). Measure of genuine multipartite entanglement with computable lower bounds. *Phys. Rev. A*, 83:062325.
- [8] Martin, A., Guerreiro, T., Tiranov, A., Designolle, S., Fröwis, F., Brunner, N., Huber, M., and Gisin, N. (2017). Quantifying photonic high-dimensional entanglement. *Phys. Rev. Lett.*, 118:110501.

- [9] Ono, T., Okamoto, R., and Takeuchi, S. (2013). An entanglement-enhanced microscope. *Nature communications*, 4.
- [10] Pan, J., Simon, C., Brukner, C., and Zeilinger, A. (2001). Entanglement purification for quantum communication. *Nature*, 410(6832):1067.
- [11] Sheng, Y.-B. and Deng, F.-G. (2010a). Deterministic entanglement purification and complete nonlocal bell-state analysis with hyperentanglement. *Physical Review A*, 81(3):032307.
- [12] Sheng, Y.-B. and Deng, F.-G. (2010b). One-step deterministic polarization-entanglement purification using spatial entanglement. *Physical Review A*, 82(4):044305.
- [13] Simon, C. and Pan, J.-W. (2002). Polarization entanglement purification using spatial entanglement. *Physical review letters*, 89(25):257901.
- [14] Strekalov, D., Pittman, T., Sergienko, A., Shih, Y., and Kwiat, P. (1996). Postselection-free energy-time entanglement. *Physical Review A*, 54(1):R1.
- [15] Tiranov, A., Designolle, S., Cruzeiro, E. Z., Lavoie, J., Brunner, N., Afzelius, M., Huber, M., and Gisin, N. (2016). Quantification of multi-dimensional photonic entanglement stored in a quantum memory based on sparse data. *arXiv preprint arXiv:1609.05033*.
- [16] Vallone, G., Dequal, D., Tomasin, M., Vedovato, F., Schiavon, M., Luceri, V., Bianco, G., and Villoresi, P. (2016). Interference at the single photon level along satellite-ground channels. *Phys. Rev. Lett.*, 116:253601.
- [17] Xie, Z., Zhong, T., Shrestha, S., Xu, X., Liang, J., Gong, Y.-X., Bienfang, J. C., Restelli, A., Shapiro, J. H., Wong, F. N., et al. (2015). Harnessing high-dimensional hyperentanglement through a biphoton frequency comb. *Nature Photonics*, 9:536–542.

Reviewer #1 (Remarks to the Author):

The authors have addressed my concerns regarding the novelty of the work. I recommend the publication of the revised draft in Nature Communications.

Reviewer #3 provided confidential remarks to the editor, supporting publication of the manuscript.

To the reviewers of NCOMMS-16-23135, ,
"Distribution of high-dimensional entanglement
via an intra-city free-space link"

May 11, 2017

Response to the Reviewers of **NCOMMS-16-23135**

We are delighted to read the reviewers' positive final assessment and thank all three reviewers for their valuable feedback. The changes made in response to your suggestions have, undoubtedly, resulted in a significantly improved final manuscript.

Yours Faithfully,

Fabian Steinlechner, Sebastian Ecker, Matthias Fink, Bo Liu,
Jessica Bavaresco, Marcus Huber, Thomas Scheidl, and Rupert Ursin